# A Novel Method for Network Design and Optimization of District Energy Systems: Considering Network Topology Planning and Pipe Diameter

Jiazheng Wu [1], Hongyun Liu [1], Yingjun Ruan [1,*], Shanshan Wang [2], Jiamin Yuan [1] and Huiming Lu [1]

1   College of Mechanical and Energy Engineering, Tongji University, Shanghai 201804, China;
    tjwujz@tongji.edu.cn (J.W.); liuhongyun4748@gmail.com (H.L.); yjm_9703@163.com (J.Y.);
    luhuiming19@163.com (H.L.)
2   College of Energy and Mechanical Engineering, Shanghai University of Electric Power,
    Shanghai 201804, China; xz_wss@163.com
*   Correspondence: ruanyj@tongji.edu.cn

**Abstract:** This paper proposes a new network topology design method that considers all the road nodes, energy stations and load centers to ensure the distribution of pipes along the road. The traditional graph theory and Prim Minimum Spanning Tree (MST) are used to simplify the map and minimize the length of the pipeline. After analyzing the limitations of the traditional network topology model, Point-to-Point (PTP), we present a new model, Energy Station-to-Load Point (ESLP). The model is optimized by minimum cost, not the shortest path. Finally, Pipe Diameter Grading (PDG) is proposed based on ESLP by solving for the pipe diameter that gives the minimum cost under different load demands in the process of optimization. The network design method is effectively applied in a case, and the results show that the path of the optimized plan is 1.88% longer than that of the pre-optimized plan, but the cost is 2.38% lower. The sensitivity analysis shows that the cost of pipeline construction, project life and electricity price all have an impact on the optimization results, and the cost of pipeline construction is the most significant. The difference between the different classifications of pipelines affects whether PDG is effective or not.

**Keywords:** district energy system; graph theory; network planning; minimum spanning tree; pipe diameter grading

## 1. Introduction

### 1.1. Research Background

With the rapid development of the social economy, energy demand is increasing year by year. Energy development and environmental issues are becoming more and more prominent, and the pressure to improve energy efficiency is mounting [1]. District Energy Systems (DES) are recommended as an effective way to improve energy efficiency and promote the use of renewable energy to maximize the efficiency of different energy supply systems [2]. A DES consists of energy stations, pipe networks and load centers [3]. The design of DES is a sophisticated assignment, which involves the integration and intersecting of different disciplines. The major tasks in DES design are the determination of the location and the capacity of the energy stations, topology planning of energy pipelines, etc. Extensive research efforts have been focused on the development of algorithms to determine the location and capacity of energy stations. However, the topology planning of energy pipelines has rarely been studied [4,5].

To minimize the power loss and cost of the energy system, the convex optimal power flow method has been applied [6,7], as well as the hybridization of metaheuristic algorithms [8]. The difference is, the former can guarantee that the found optimum is the global optimum, while the latter cannot. In another study, J.H. L. and C.C. S. proposed a comprehensive energy system planning and operation framework, which set technical index,

economic index, and environmental index as objective functions, and considered social factors, market factors and safety margin as constraint conditions, unfortunately, without a mathematical model to run it [9]. Based on the planning example of a park and the concept of multi-scene planning, L. C. and J. Z. analyzed the subsystem planning of the integrated energy station and the benefits of the multi-energy complementary system [10]. The hybrid integer nonlinear algorithm showed excellent performance in dealing with the optimization of renewable energy system, whose efficiency was low during the voltage recovery process because of the rapid response of the renewable energy system [11]. The planning of a regional integrated energy system based on Combined Cooling, Heating and Power (CCHP) system and heat network was studied, and the planning of capacity optimization of multi-regional CCHP system was analyzed. However, the heat network model based on mixed integer linear programming is so simplified, while ignoring temperature and pressure changes, that it obtained a result that required further discussion [12]. It should be noted that the abovementioned studies only paid attention to the size and location of energy stations in the optimization, without considering the interplay between energy stations and energy transmission networks.

The transmission network is also an important part of the district energy system, and has a strong influence on the operational performance of DES [13]. Major factors in the network design include pipeline diameter [14,15], pipeline topology [16], field heat loss [17], mass flow, and thermal conductivity distribution [18]. A novel methodology involving both kernel density estimation and the shortest path method was proposed to optimize the transmission network [19]. In their study, the transmission path was optimized using an energy distance function [16], but the pipelines were distributed without considering the actual road distribution. The impact of the quantity and layout of the heating network on the carbon footprint of the system was discussed [20]. Based on the energy flow balance constraint and heat network characteristics, the layout planning model of the energy station supply pipeline was established, and the resolution process was simplified using hybrid mutation particle swarm optimization algorithm [21], but the results cannot be proved to be optimal in the process of simplification. The modified non-dominated sorting genetic algorithm II, with the goal of cost minimization of the distribution network and profit maximization of the microgrid [22], was only applicable for resolving the power in DES. The graph theory technique made it possible to quickly find the optimal solution, but the load points were connected directly, ignoring the road distribution. An artificial neural network-based reinforcement learning algorithm was proposed to optimize the multi-energy management-based energy routing design. This method is suitable for studying the energy exchange between load points, but is of little help for pipeline topologies in physical space [23]. Additionally, due to the "no free lunch" theorem of mathematical optimization, the selection of solvers depends on the optimization task. There is a most appropriate metaheuristic solver for a specific application [24]. Anyway, the design and optimization of the network in DES should meet the goals of maximum energy efficiency, system stability [25] and renewable energy share [26], at the lowest economic costs [15,20,27], energy losses [28] and carbon emissions [20]. However, the state-of-the-art mathematical optimization model and calculation algorithm are relatively complex, and the actual situation of pipeline distribution along the road has rarely been considered in previous studies. Graph theory is the most direct and effective method for topology design of pipeline networks, and we improve it on the basis of ignoring the road distribution in traditional application methods. Therefore, a simplified pipe network design optimization model that considers the pipeline distribution along the road was developed and solved using a minimum spanning tree algorithm.

### 1.2. Motivation

Most of the studies on the pipe network design of regional energy systems directly connect the load points without taking into account the fact that, in practice, pipelines should be laid along the road, at present. Therefore, it is necessary to adopt a network

design method for district energy systems that fully considers the distribution of pipelines along the road. Additionally, the minimum spanning tree method has been widely used to optimize the network topology, with the minimum total path as the objective function. How to couple the two main elements in pipe network design, pipe diameter ($d_i$) and pipe length ($l_i$), while taking the minimum annual average cost as the optimization goal, is the difficulty in the current research.

*1.3. Contribution*

In this paper, a method for designing the network of DES considering pipeline distribution along the road is presented. According to the principle of Pipe Diameter Grading (PDG), a suitable pipe diameter is selected according to the load of the studied area and the position of the pipe segment. Then, the pipe diameter is coupled with the pipe length, and the minimum annual cost of the pipe network is set as the optimization goal.

The innovations of this paper include: (a) The limitations of the traditional method (Point-to-Point, PTP) of building connection graph are analyzed and an improved one (Energy Station-to-Load Points, ESLP) is proposed. (b) On the basis of ESLP, we propose a novel method of network design and optimization.

*1.4. Framework of the Paper*

Figure 1 shows the outline of this paper, which comprises two parts. The first part discusses the planning of the pipeline networks connection by comparing two pipe network connection models: Point-to-Point Model (PTP) and Energy Station-to-Load Points Model (ESLP). The second part is focused on the optimization of the Pipe Diameter Grading (PDG) based on the ESLP model.

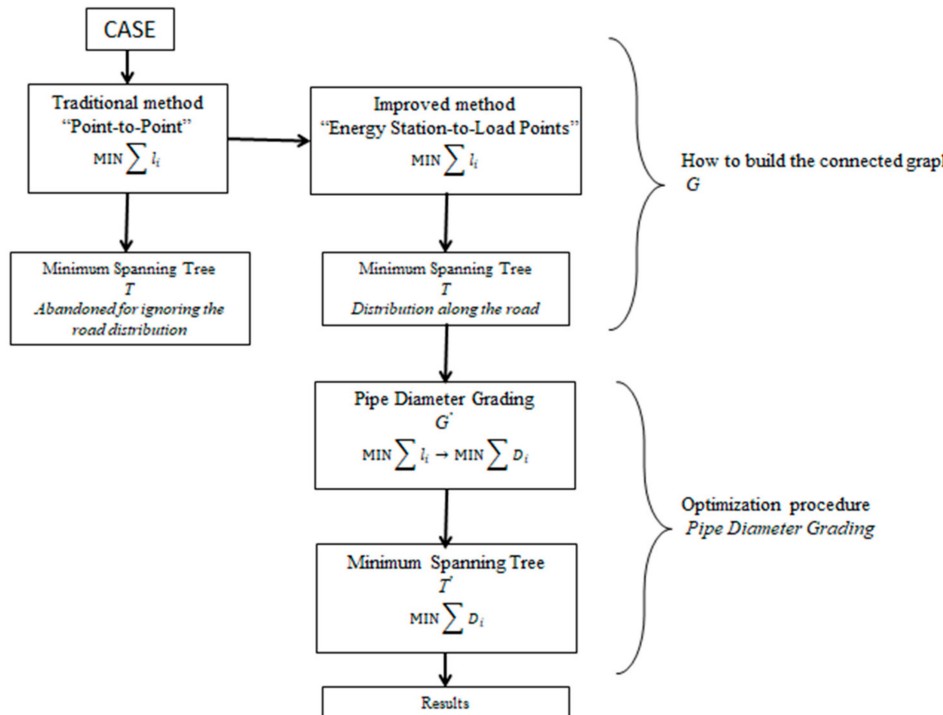

**Figure 1.** The main research content of this paper.

## 2. Minimum Spanning Tree

*2.1. Definition*

Minimum spanning tree is often used to solve practical problems of applying graph theory knowledge. It is widely used in computer science, mathematics, and data structure fields. The Kruskal algorithm and the Prim algorithm are the two most used algorithms in the minimum spanning tree. The minimum spanning tree is defined as [29] follows:

Let G = (V, E) denote a connected graph, where V and E represents the vertex set V = {$v_1$, $v_2$,..., $v_n$} and the edge set E = {1, 2,..., m}, respectively. A spanning tree T = T (V, S) is defined as a connected acyclic subgraph of G that includes all vertices. Obviously, S is a subset of E. To simplify the model, we denote a spanning tree T by its edge set S in this paper. represents the edge connecting vertex $u$ to vertex $v$, and $w$ represents the weight of this edge. A spanning tree $T^0$ is said to be a minimum spanning tree if Equation (1):

$$\sum_{(u,\ v)\in T^0} w(u,\ v) \leq \sum_{(u,\ v)\in T} w(u,\ v), \tag{1}$$

holds for any spanning tree T.

A graph G with 4 vertices and 6 edges is shown in Figure 2. When E = (1, 3, 4, 2, 2, 3), the unique minimum spanning tree is $T^0$ = {AB, AD, AC} with $\sum_{(u,\ v)\in T^0} w(u,\ v) = 1 + 2 + 2 = 5$.

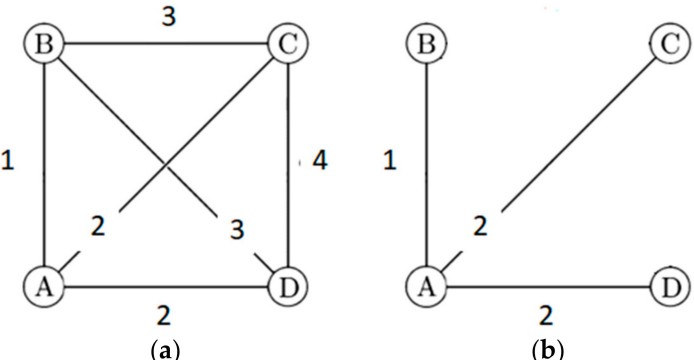

**Figure 2.** (**a**) Connected graph G and (**b**) minimum spanning tree for example.

According to the principle of minimum spanning tree, if the studied area is simplified to a connected graph, where the distance relation is the weight of the connected graph, then the calculated result is the topological project with the shortest total pipeline length. It can be inferred that the minimum spanning tree algorithm is suitable for the design and research of pipe network in DES. Prim algorithm is employed for resolution of the model, and the calculation operated on MATLAB R2018a software. The calculation steps are briefly described as follows:

(1) Input: A weighted connected graph, G = (V, E). The set of vertices is V, the set of edges is E;
(2) Initialization: $V_{new}$ = {x}, x is any node (starting point) in set V. $E_{new}$ = {} is an empty set;
(3) Repeat the following until $V_{new}$ = V:
    i. Select the edge ($u$, $v$) with the smallest weight in set E, where u is an element in set $V_{new}$ and $v$ is not in the set $V_{new}$, and $v \in$ V;
    ii. Add v to set $V_{new}$, and add edge ($u$, $v$) to set $E_{new}$;
(4) Output: The collections $V_{new}$ and $E_{new}$ are used to describe the resulting minimum spanning tree.

## 2.2. How to Build the Connected Graph "G"

This paper presents a case study in Zhejiang Province, China. The studied area occupies an area of 1.43 square kilometers and houses various types of buildings. The division and numbering of load areas are shown in Figure 3. Before applying the method of MST, it is necessary to preconnect all paved pipeline sections in the study area. In this paper, the pipeline network design optimization process of DES is constrained by the pipeline distribution along the road. Therefore, the pretreatment of the studied area is different from the common methods. First, the studied area is divided into different blocks according to the road distribution, and each block is treated as a load area. Additionally, to describe the topology model between source and load within the pipe network, the load area is

simplified to a load center that is located in a place within the load area. Second, besides configuration of Energy Station (star) and Load Points (dots), Road Nodes (squares) and Load-Road Nodes (triangles), which represent the intersections of load points and nearby roads, should be taken into account to ensure that pipelines are laid along the road, as shown in Figure 3. It should be noted that the construction of the hot water pipe between the end-user load point and the road is complicated in practice and generally not considered in the study. It has been simplified in this article. In addition to this, Energy Station, which has little relation with the design method of pipeline network studied in this paper, has also been simplified as a heating plant.

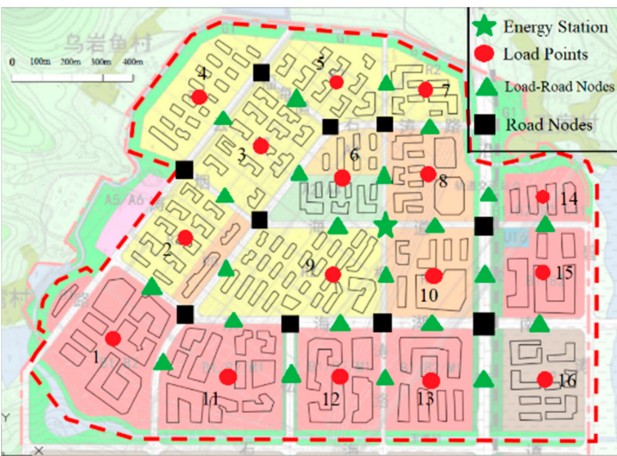

**Figure 3.** Preprocessing of the study area.

Because the pipeline must be distributed along the road, all pipeline network connections follow two principles:

(1) All load points (dots) must be connected to the adjacent load-road nodes (triangles).
(2) Any adjacent two points connected, but the pipe where two ends are not load points (dots) must be located on the road. The pipeline network graph was created using two methods, PTP and ESLP.

### 2.2.1. Traditional Method, PTP

A conventional configuration of pipeline network is based on a Point-to-Point model (PTP), which connects any two adjacent points in the studied area, ignoring their own properties, and whether they are load points or energy stations or road nodes, as shown in Figure 4. The digital adjacency matrix is generated after constructing the pre-connected grid, and then the optimal topology scheme is solved using the minimum spanning tree algorithm. However, the results show that the solution obtained by the PTP model has a defect, and this method of building connected graph needs to be improved.

The pipeline grid was arranged configured according to the PTP model for the studied area, (Figure 5). The corresponding adjacency matrix was constructed, and the minimum spanning tree was solved using the Prim algorithm.

It should be explained that the minimum spanning tree calculates all the nodes in the model, including both useful and inactive tips. In this study, the inactive tips are rejected manually from the model after obtaining the minimum spanning tree. The inactive tips are selected based on two conditions: (1) one end of the tip is a leaf node, that is, a terminal node; and (2) the terminal node is not a Load Point.

The resulting DES pipeline topology after rejecting inactive tips is shown in Figure 6. It can be seen that only a small portion of the pipelines are laid along the road, and most of pipelines are constructed by simply connecting the load points directly, without considering the road nodes. If only two types of nodes, energy station (star) and load point (dots), were taken as the research objects at the beginning, and road nodes were not taken into

account, the results obtained would be similar to this scheme. It can be inferred that the design of the pipeline ignored the existence of roads and road nodes with this method, and this needs to be improved. The reason for this is that the four types of nodes were treated and dealt with uniformly at the beginning. No matter what kind of points they were, they were all "point-to-point". Therefore, the conventional PTP method is no longer suitable for the study in this paper. To solve this problem, it is necessary to treat the connected graph according to the properties of different nodes.

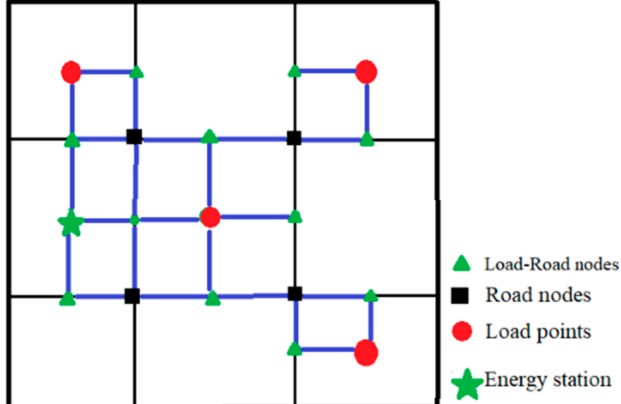

**Figure 4.** A schematic of the connected graph G using the Point-to-Point model.

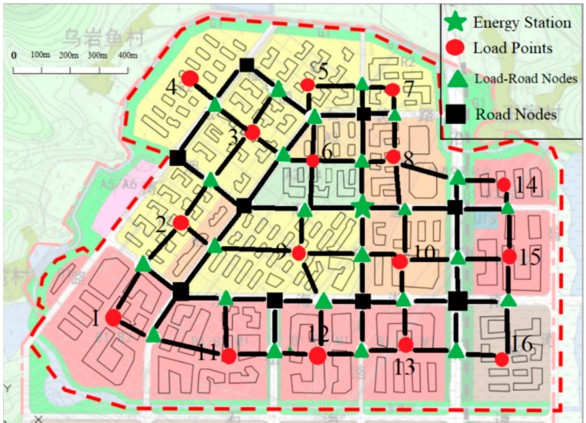

**Figure 5.** Connected graph G of the study area using the Point-to-Point model.

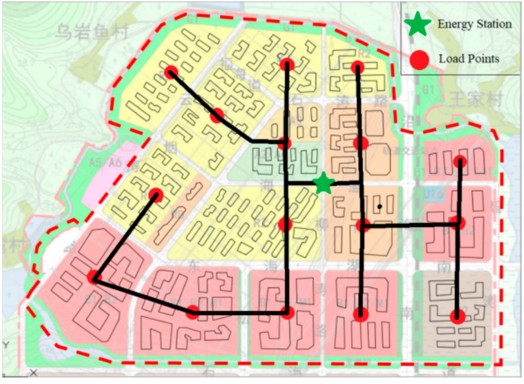

**Figure 6.** Network topology plan guided by the Point-to-Point model.

### 2.2.2. Improved Method, ESLP

To overcome the defects in PTP, an Energy Station-to-Load Points model is developed. The model does not connect all the adjacent points randomly, and conforms to the following rules:

a.　Select Energy Station and Load Points according to their own characteristics
b.　Only the path that starts from the Energy Station and ends at a Load Point is calculated. The reverse path that satisfies Equation (2) is not calculated, to simplify the calculation.

$$\cos V_{tp} V_{sp} L \leq 0, \tag{2}$$

where $V_{tp}$ represents the end point of the line segment, $V_{sp}$ represents the starting point of the line segment, and $L$ represents the target Load Point.

c.　Repeat the above steps until all Load Points have been connected.

For easy solution, more constraints are added to the ESLP model as follows:

(1)　Load points (dots) can only be connected to adjacent load-road nodes (triangles).
(2)　Any two adjacent points can be connected, but pipe that does not end at load points (dots) must be located on the road.
(3)　Load points (dots) can only be set as the end point of a line segment

Figure 7 shows the ESLP diagram. This configuration guarantees that all line segments are connected to load points in one direction, thus ensuring that each load point is the last terminal of the final network topology. Compared with the traditional PTP model, the ESLP model has two benefits. First, it guarantees that all the pipelines are distributed along the road. Second, the load points are designated as the terminal of the network topology, allowing for easy solution in the subsequent optimization process.

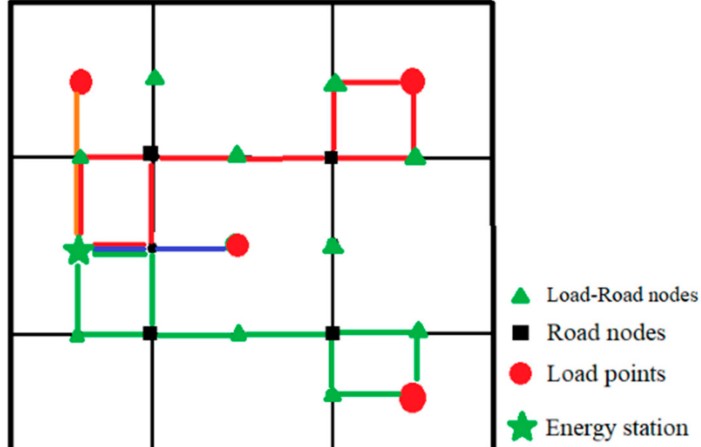

**Figure 7.** A schematic of connection graph G by Energy Station-to-Load Points model.

The modeling procedure of ESLP mainly consists of the generation of a grid connected graph, the construction of the digital adjacency matrix, and the optimization of topology using the minimum spanning tree algorithm. The subsequent optimization of pipe diameter selection is based on the ESLP model.

ESLP is a different type of pipe network connected graph construction model. It divides nodes (which include load nodes, road-load nodes, load points, and energy stations) according to their own characteristics, avoiding the problems caused by unconstrained pipeline connections. Instead of connecting any two adjacent points, ESLP first selects energy stations and load points, and connects energy stations to all load points, respectively and independently.

After classifying the four categories of nodes, new results are obtained according to the principles mentioned above, as shown in Figure 8, when constructing a new network connected graph. The new pipe network design scheme is shown in Figure 9.

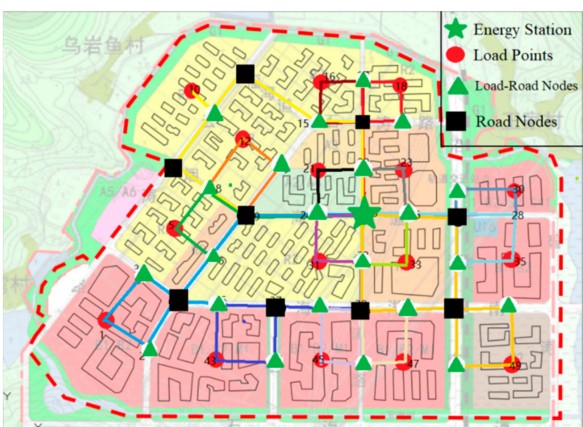

**Figure 8.** Connected graph G of the study area using the Energy Station-to-Load Points model.

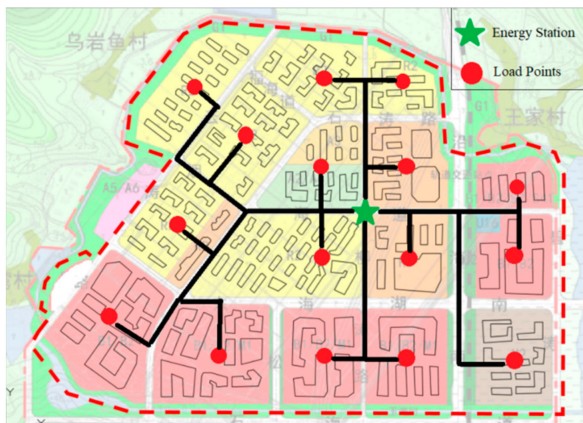

**Figure 9.** Network topology plan guided by ESLP.

Obviously, ESPL ensures that pipes are laid along the road. However, previous studies have been conducted on the basis of the relative position relationship between nodes. The resulting network topology scheme also targets the optimal path, that is, the shortest pipe length, which is inconsistent with the cost optimization goal of network design in DES. The economic cost is related to the pipe diameter and pipe length, which will be discussed in Section 3.2.

## 3. Optimization Procedure by PDG

### 3.1. Network Economic Cost

The average annual cost of the pipe network (*D*) can be expressed as Equation (3) [16,19,30]:

$$D = C_0^P \frac{r(1+r)^L}{(1+r)^L - 1} + C_w^P + C_m^P + C_f^P, \tag{3}$$

where $C_0^P$ represents the initial investment in pipeline construction; $C_w^P$ represents the heat loss cost of the pipe; $C_m^P$ represents the depreciation and maintenance cost of the pipe;

and $C_f^P$ represents the operating cost of the pump. $r$ is the annual interest rate and $L$ is the life of the project. Additionally, Equation (4):

$$C_o^P = \sum_{i=1}^{n} C(d_i) l_i, \tag{4}$$

where $C(d_i)$ represents the construction cost of pipe section $i$ per meter of pipe length, including supply and return water pipeline; $d_i$ represents the pipe diameter of section $i$; $l_i$ represents the pipe length of segment $i$. The heat loss costs are calculated by Equation (5):

$$C_w^P = \sum_{i=1}^{n} \frac{4\pi\lambda t_c l_i T_p C_e}{ln\frac{d_i + 2\delta_i + \delta_{ti}}{d_i + 2\delta_i} \times COP \times 10^7}, \tag{5}$$

where $\lambda$ is the thermal conductivity coefficient of thermal insulation material coated on the pipeline; $t_c$ is the average temperature difference between inside and outside the water conveyance pipeline; $T_p$ is the annual maximum working time of circulating pump; $C_e$ is the electricity price; $\delta_i$ is the thickness of the pipe; $\delta_{ti}$ is the thickness of the insulation layer; COP is the energy efficient ratio of the system. The depreciation and maintenance costs are calculated by Equation (6):

$$C_m^P = \left(\mu_1^P + \mu_2^P\right) C_o^P, \tag{6}$$

where $\mu_1^P$ represents the equipment depreciation rate, and $\mu_2^P$ represents the maintenance cost proportional coefficient. The pumping costs are calculated by Equation (7):

$$C_f^P = \sum_{i=1}^{n} 2.78 \times 10^{-7} \frac{Q_i \cdot H_{Pi}}{\rho \cdot \eta_P} \cdot T_p C_e, \tag{7}$$

where $C_f^P$ represents the annual operating cost of the circulating pump; $Q_i$ represents the calculated flow rate of pipe segment $i$; $H_{Pi}$ represents the head of circulating pump; $\rho$ represents the density of the heat vector and water is used as the heat vector in this text. $\eta_P$ represents the motor efficiency of the circulating pump; $T_p$ represents the number of working hours of circulating water pump calculated in the next year; $C_e$ is the price of electricity.

If the pump power set in each branch pipe segment is fully used to overcome the flow resistance, then the head value of circulating pump in the pipe segment can be expressed by the resistance loss in the pipe segment. The resistance loss of pipe section consists of two parts: the resistance loss due to friction and the local resistance loss of the fluid, as expressed in Equation (8):

$$H_{Pi} = \Delta P_i = \Delta P_{mi} + \Delta P_{ji}, \tag{8}$$

where $\Delta P_{mi}$ represents the resistance loss along the pipe segment i; $\Delta P_{ji}$ represents the local resistance loss of section $i$.

The resistance loss along the pipe is caused by the friction between the fluid and the wall surface of the pipe in the straight pipe. Its calculation formula is Equation (9), as follows:

$$\Delta P_{mi} = \gamma \cdot \frac{l_i}{d_i} \cdot \frac{\bar{v}^2 \rho}{2}, \tag{9}$$

where $\bar{v}$ is the velocity of thermal media, $\gamma$ is the resistance coefficient along the inner wall of the pipe, which can be determined by schifflinson formula, Equation (10):

$$\gamma = 0.11 \left(\frac{K}{d_i}\right)^{0.25}, \tag{10}$$

where $K$ represents equivalent roughness of pipe inner wall, which can be determined directly according to the design specification of urban heating pipe network. For the section with water as the heat medium and steel pipe as the pipe material, $K$ equals $5 \times 10^{-4}$ m.

The local resistance loss is caused when the fluid medium passes through the pipe parts such as bending pipe or expanding and shrinking. The local resistance loss is dependent on the geometries of pipeline segments and the calculation is complicated. Under normal conditions, the local resistance loss of thermal pipe network can be converted represented by $l_i$, the corresponding length of pipe section, and the local resistance coefficient, as the equivalent length of local resistance loss, $l_{ji}$ and $\Delta P_{ji}$ are calculated using Equations (11) and (12):

$$l_{ji} = \alpha \cdot l_i, \tag{11}$$

$$\Delta P_{ji} = \alpha \cdot \Delta P_{mi} \tag{12}$$

In the formula, $\alpha$ represents the local resistance coefficient. According to the design specification of urban heating pipe network, $\alpha$ ($0.15 < \alpha < 0.2$) is set as 0.2 in this paper.

### 3.2. Constraint Conditions

In the pipe network design of the district energy system, the diameters of the pipe are not consistent, and this determines multiple factors, such as volume rate of flow and velocity of flow. These factors follow Equation (13):

$$Q_i \leq \frac{1}{4}\pi d_i^{re2}\overline{v}_i, \tag{13}$$

where $Q_i$ represents the design flow of the calculation area; $d_i^{re}$ represents the calculated reference diameter; $\overline{v}_i$ represents the velocity of the thermal media. In practical engineering construction, the pipe diameter of thermal network is not a continuous variable, but rather a discrete variable with a nominal value. According to the relevant engineering estimation indexes, the selection of pipe diameter should be performed with reference to Equation (14):

$$d_i^{st} \leq d_i^{re} \leq d_i = d_i^{st+1}, \tag{14}$$

where $d_i^{st}$ and $d_i^{st+1}$ represent two adjacent standard pipe diameters in the engineering standard, and $d_i$ is the final selected pipe diameter of the pipe network design.

The weight $w(u, v)$ of the edge $(u, v)$ in the connected graph G is the distance between points $u$ and $v$. Therefore, the optimization goal of the minimum spanning tree method is the minimum pipe length, Equation (15):

$$\text{MIN} \sum w(u, v) = \text{MIN} \sum l_i, \tag{15}$$

However, in the actual project, the design optimization goal is the minimum cost, Equation (16):

$$\text{MIN } D, \tag{16}$$

It can be seen from Section 2.2 that $D$ is not only affected by pipe length $l_i$, but also pipe diameter $d_i$, as per Equation (17):

$$D = f(l_i, d_i) = \left(\mu_1^P + \mu_2^P + \frac{r(1+r)^L}{(1+r)^L - 1}\right) \times \sum_{i=1}^{n} C(d_i)l_i + \sum_{i=1}^{n} \frac{4\pi\lambda t_c l_i T_p C_e}{ln\frac{d_i + 2\delta_i + \delta_{ti}}{d_i + 2\delta_i} \times COP \times 10^7} + \sum_{i=1}^{n} 1.8 \times 10^{-8} \frac{Q_i l_i T_p C_e \overline{v}^2}{d_i \eta_P}\left(\frac{K}{d_i}\right)^{0.25}, \tag{17}$$

This is not a simple numerical optimization question, because the topology of the pipe network and pipe dimensions must be considered at the same time. The design optimization should be applied in a multidisciplinary manner. Combined with the method in this paper, the minimum spanning tree, the weight $w(u, v)$ of edge in connected graph G should be determined by distance $l_i$ correction for the cost $D_i$, the optimization goal should be determined by MIN $\sum l_i$ correction for MIN $\sum D_i$. To achieve this goal, the Pipe Diameter

Grading theory (PDG) is proposed. Two fundamental assumptions in the PDG theory are: (A) Network topology diagram is a nonlinear hierarchical tree structure. (B) In this structure, the root node is the location of the energy station, and there is no precursor node. (C) All leaf nodes are load point positions, and are the last terminal, with no subsequent nodes. Each of the remaining nodes can be followed by one or more nodes. The schematic diagram of PDG is shown in Figure 10. The $D_i$ of any pipe segment can be expressed as Equation (18):

$$D_i = \left( \mu_1^P + \mu_2^P + \frac{r(1+r)^L}{(1+r)^L - 1} \right) \times C(d_i)l_i + \frac{4\pi \lambda t_c l_i T_p C_e}{ln \frac{d_i + 2\delta_i + \delta_{ti}}{d_i + 2\delta_i} \times COP \times 10^7} + 1.8 \times 10^{-8} \frac{Q_i l_i T_p C_e \overline{v}^2}{d_i \eta_P} \left( \frac{K}{d_i} \right)^{0.25}, \quad (18)$$

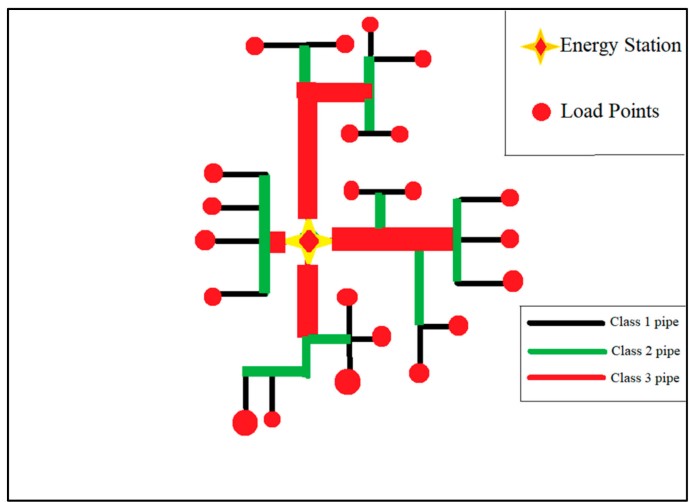

**Figure 10.** Pipe Diameter Grading when the classification of the pipe is 3.

Pipe Diameter Grading means that different pipelines are graded according to their position relative to energy stations and load points. The pipeline levels are classified as follows:

Class 1 pipeline: From the load point to the node of another or more class $N(N \geq 1)$ pipelines, where the pipe is called class 1 pipeline, shown as black line in Figure 10;

Class 2 pipeline: Starting from nodes of two or more Class 1 pipelines to nodes of another or more Class $N(N \geq 2)$ pipelines, where the pipelines are called stage 2 pipelines, shown as a green line in Figure 10;

. . .

Class $i$ pipeline: Starts from the nodes of two or more Class $(i - 1)$ pipelines to nodes of another or more Class $N$ ($N \geq$ i) pipelines, where the pipelines are called class $i$ pipelines.

Pipe diameter grading is based on flow balance, as expressed in Equation (19):

$$\dot{Q}^N = \sum_{i=1}^n \dot{Q}_i^{N-1} + \sum_{i=1}^n \dot{Q}_i^{N-2} + \ldots + \sum_{i=1}^n \dot{Q}_i^1, \quad (19)$$

where $\dot{Q}^N$ refers to the Class-$N$ pipe flow rate to the same node; Equation (20):

$$\dot{Q}^N \geq \dot{Q}_i^m, \quad (20)$$

By classifying the pipes at different levels, different diameter characteristics can be assigned to them, so that all edge weights $w(u, v)$ can be changed from distance $l_i$ to cost $D_i$ in the connected graph G. Then, the Prim algorithm of the minimum spanning tree can be used to find the optimal scheme of the minimum cost for further optimization.

### 3.3. Pipe Diameter Grading

The design flow of each load area is shown in Table 1.

**Table 1.** The design flow of each block.

| Load Area ID | Design Flow (kg/h) | Load Area ID | Design Flow (kg/h) |
| :---: | :---: | :---: | :---: |
| 1 | 12,390 | 9 | 13,440 |
| 2 | 8050 | 10 | 11,020 |
| 3 | 8420 | 11 | 29,480 |
| 4 | 7770 | 12 | 3910 |
| 5 | 8520 | 13 | 30,510 |
| 6 | 14,090 | 14 | 17,100 |
| 7 | 4360 | 15 | 26,660 |
| 8 | 17,820 | 16 | 14,110 |

In the network design of DES, the choice of pipe diameter depends on the network topology. On the other hand, under the optimization goal of the minimum cost, the topological scheme of the pipe network is also affected by pipe diameter. The pipeline length and the pipe diameter are highly coupled and mutually influential. To solve this problem, we propose a Pipe Diameter Grading (PDG) method, which grades different pipe segments according to their relative positions with respects to energy station and load points, and assigns different characteristic pipe diameters accordingly, so as to realize the transition of edge weights in connected graphs from distance to cost.

It can be observed from Figure 10 that, according to the PDG, the first identified Class1 pipeline is directly connected to the load point, and all the load points are connected to the Class1 pipeline, and the pipe diameter of the Class1 pipeline is directly determined by the design flow rate of the connected load point. Therefore, it is necessary to clarify "which is the Class1 pipeline" and "which load point is connected to the Class1 pipeline". The three principles of ESLP mentioned in Section 2.2.2 are intended to solve this problem, which is why it is said that "the load point serves as the terminal of the network topology is the basis of PDG".

The network topology scheme in Figure 9 is used to determine the pipe diameters of each pipe segment according to the pipe diameter selection rules in Section 3.2, as shown in Figure 11. The parameters for cost estimation are shown in Table 2 [17,20,30].

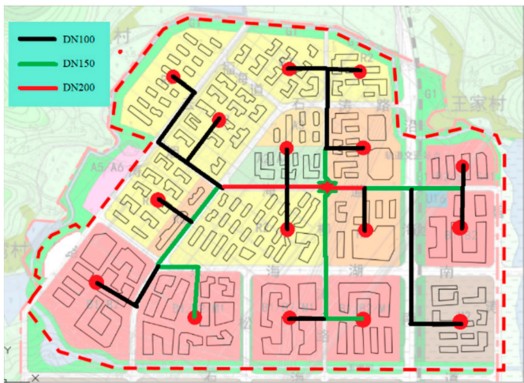

**Figure 11.** The pipeline network scheme with the minimum path as the optimization objective.

**Table 2.** Table of simulation references.

| Parameters | Values | Units | Parameters | Values | Units |
|:---:|:---:|:---:|:---:|:---:|:---:|
| $r$ | 6% | | $\delta_{DN100}$ | 2.5 | mm |
| $C(d_{DN100})$ | 609.41 | RMB/m | $\delta_{DN150}$ | 3.5 | mm |
| $C(d_{DN150})$ | 864.46 | RMB/m | $\delta_{DN200}$ | 4.5 | mm |
| $C(d_{DN200})$ | 1206.26 | RMB/m | $\delta_{tDN100}$ | 25 | mm |
| $\lambda$ | 0.06 | W/(m·K) | $\delta_{tDN150}$ | 30 | mm |
| $L$ | 20 | year | $\delta_{tDN200}$ | 40 | mm |
| $t_c$ | 10 | °C | $\mu_1^P$ | 0.01 | |
| $T_p$ | 3360 | h | $\mu_2^P$ | 0.015 | |
| $Ce$ | 0.6 | RMB/(kWh) | $\rho$ | 1000 | kg/m$^3$ |
| COP | 4.2 | | $\eta_P$ | 0.6 | |
| $\bar{v}$ | 0.8~1.2 | m/s | $\alpha$ | 0.2 | |
| $K$ | $5 \times 10^{-4}$ | m | | | |

It should be noted that the cost is estimated based on the minimum pipeline length. The next step is to study and analyze the target area, classify the different pipe segments, and perform further optimization with minimum cost as the target.

## 4. Results and Discussion

### 4.1. Results

According to the calculation and analysis of multiple iterations, the diameter of the pipeline in the targeted area is distributed within the interval of DN100~DN200. The pipeline segments in the area can be divided into three levels, with three different nominal values (DN100, DN150 and DN200). The results of pipe diameter grading are shown in Figure 12.

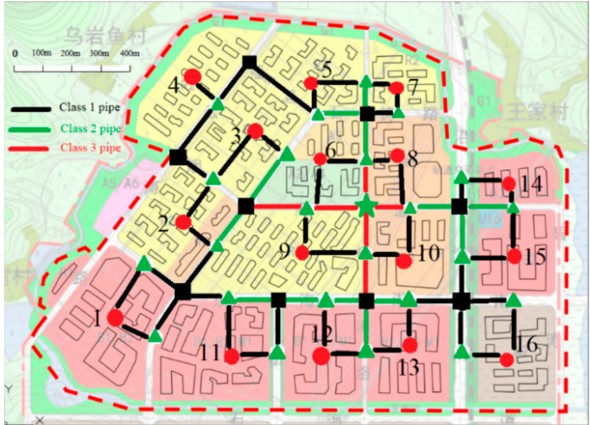

**Figure 12.** The result of the Pipe Diameter Grading in the study area.

In this way, the weight of any pipe segment in the target area can be updated to $D_i$ according to Equation (18), and the original problem is transformed into a minimum spanning tree problem with network annual cost as the weight. The Prim algorithm is used to solve for the network topology that gives the minimum annual cost. It should be noted that in the process of pipe diameter grading, the pipe diameter is regarded as a fuzzy variable, denoted as "fuzzy pipe diameter", not the pipe segment diameter in the final scheme. There are two reasons for this:

First, the velocity of the fluid in the pipe is a variable which changes positively as a function of pipe diameter. According to the relationship between the flow rate and the pipe diameter, the selection of the appropriate pipe diameter will fluctuate within a certain range. In addition, since pipe diameters in the studied area do not vary significantly, it can be described by a fuzzy variable with permitted fluctuation.

After the topology scheme of the pipe network is calculated, the diameters of each pipe segment are determined according to Equations (13) and (14), described in Section 3.2. Afterwards, the "fuzzy pipe diameter" estimated by the pipe diameter grading is adjusted accordingly, to iterate for the optimal solution.

It can be found that the method of pipe diameter grading is feasible, and the final pipe network construction plan with minimum annual cost as the optimization goal is shown in Figure 13. The optimization results with the shortest path and minimum cost are compared in Figure 14. Although the latter increases the total pipe length by 1.88%, the cost decreases by 2.38%. It can be concluded that the method introduced in this paper is efficient and adaptable for optimizing the design of district energy system pipe networks.

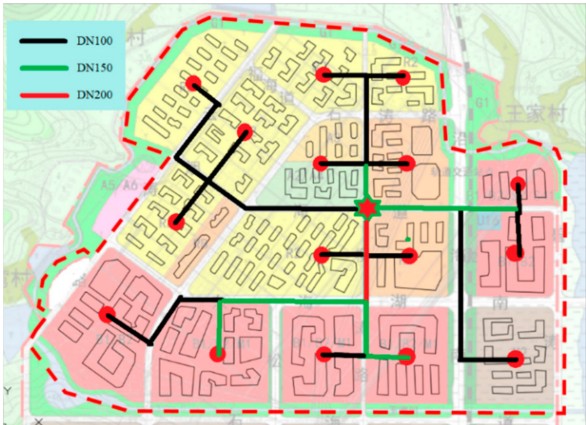

**Figure 13.** The network scheme with minimum cost as the objective of optimization.

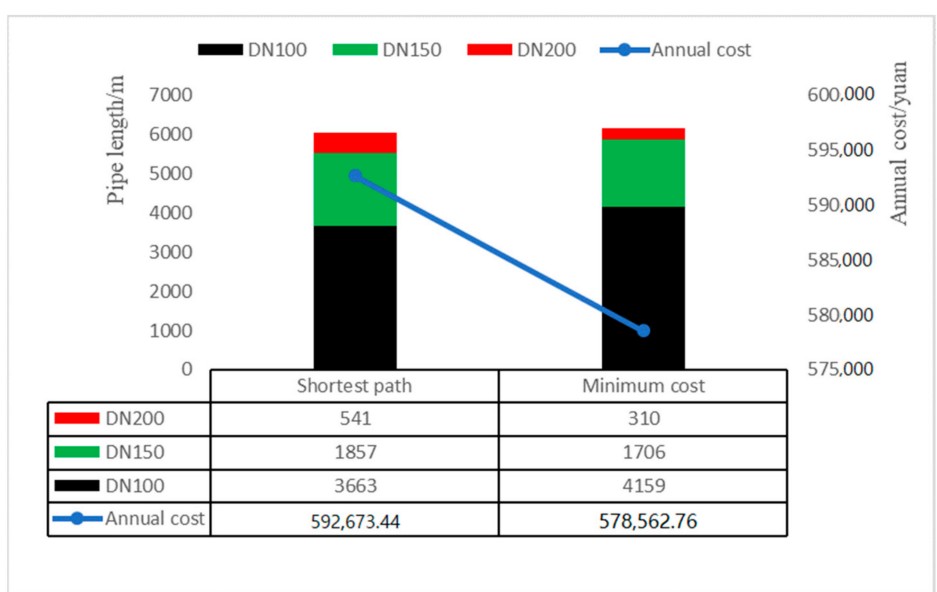

| | Shortest path | Minimum cost |
|---|---|---|
| DN200 | 541 | 310 |
| DN150 | 1857 | 1706 |
| DN100 | 3663 | 4159 |
| Annual cost | 592,673.44 | 578,562.76 |

**Figure 14.** Comparison of two schemes before and after optimization.

### 4.2. Further Discussion

In the course of the study, we found that project life ($L$), pipeline construction costs ($C(d_i)$), and electricity prices ($C_e$) were the main factors determining the total cost. To ensure the flexibility and adaptability of this design method, we carried out sensitivity analysis of the upper and lower 25% of the above parameters. The comparison with the scheme in Figure 13 is shown in Table 3. It can be found that has the greatest impact on the results, as high as 21.33%, which seems incredible. This is because we use it not only for the initial investment cost, but also for depreciation and maintenance. The second is $L$. It can be seen

from the formula of average annual cost that the impact on the results will decrease with increasing values of *L*. has the least impact, and only determines the cost of heat losses and pumping costs.

**Table 3.** The effect of changing parameters on the results.

| Parameters | Values | Units | Result (RMB) | Rate of Change of Parameter | Rate of Change of Result |
|---|---|---|---|---|---|
| $L$ | 15 | year | 649,075.22 | −25% | 12.19% |
| $L$ | 25 | year | 539,796.42 | +25% | −6.70% |
| $C(d_i)$ | - | RMB/m | 455,193.48 | −25% | −21.32% |
| $C(d_i)$ | - | RMB/m | 701,952.06 | +25% | 21.33% |
| $C_e$ | 0.45 | RMB/(kWh) | 561,882.10 | −25% | −2.88% |
| $C_e$ | 0.75 | RMB/(kWh) | 595,263.44 | +25% | 2.89% |
| $C(d_{DN100})$ | 761.76 | RMB/m | 650,205.82 | +25% | 12.38% |
| $C(d_{DN150})$ | 1080.58 | RMB/m | 619,871.07 | +25% | 7.14% |
| $C(d_{DN200})$ | 1507.83 | RMB/m | 589,042.24 | +25% | 1.81% |
| $C(d_{DN100})$ | 810 | RMB/m | 672,862.29 | +33.1% | 16.38% |
| $C(d_{DN200})$ | 750 | RMB/m | 561,895.12 | −37.8% | −2.81% |

The essence of the design optimization method (Pipe Diameter Grading) proposed in this paper is to minimize the total cost of the project on the basis of meeting the requirements of the system by flexibly using the differences between the pipelines of each classification. In other words, the core of this method is the grading of pipelines into different classifications, as well as there being sufficient differences between different classifications. If the differences between the different classifications of pipelines are too small, the PDG will fail. In this regard, we selected the parameter $C(d_i)$ with the largest influence weight ratio, and tried to narrow the gap of $C(d_{DN100})$, $C(d_{DN150})$ and $C(d_{DN200})$. The results show that when $C(d_{DN100})$ is increased by 33.1% or $C(d_{DN200})$ is decreased by 37.8%, the optimized topology changes. The final result (shown in Figure 15) is exactly the same as that (Figure 11) obtained by ESLP before introducing PDG. In both cases, $C(d_{DN100})$ and $C(d_{DN200})$ are almost identical to $C(d_{DN150})$. This indicates that, in order to ensure the effectiveness of PDG, it is necessary to ensure that the characteristics of each pipeline classification are sufficiently different.

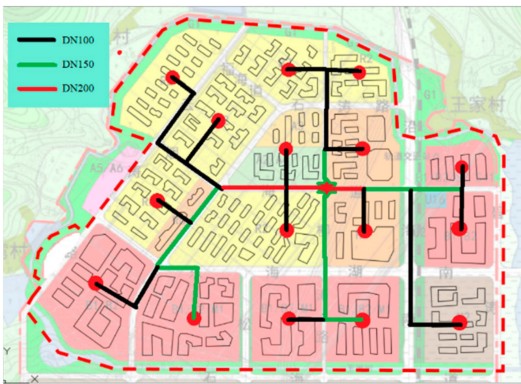

**Figure 15.** Optimized result when DN100 is increased by 33.1% or DN200 is decreased by 37.8%.

*4.3. Limitations and Future Work*

Due to the simple structure of the case in this paper, the number of pipe diameter grades is small, and there are fewer nominal diameters and pipe segments to choose from, so the results before and after optimization are not obvious. In addition, the number of iterative steps in this case is restricted, and the iterative objective function would increase after the calculation of the scheme in Figure 13. Therefore, the convergence of the algorithm introduced in this paper needs to be further verified using more complex

cases. Additionally, part loads have an impact on optimization study. In reality, loads fluctuate and change continuously over time. Although we found that the part load had an impact no greater than 10% on the results, it is valuable to take part loads into account to verify the effectiveness of the proposed design method after we obtain detailed load data in the future.

## 5. Conclusions

Most existing studies only consider the relative positions between the source and load, without considering road planning. Targeting this problem, a new design method is proposed for pipeline networks based on Graph Theory and solved using the Prim algorithm for the minimum spanning tree. The innovation of this paper is that, in addition to load points and energy station, load nodes and road-load nodes are also included in the point set of connected graphs, and two construction modes of connected graphs are compared and analyzed: PTP and ESLP. The limitations of the former are analyzed, and the improved latter is proposed. Both the theoretical analysis and calculation results show that ESLP can ensure the distribution of pipelines strictly along the road. In addition, on the basis of ESLP, the following optimization is carried out.

The pipe network design of district energy system is a problem of high complexity, strong coupling and great difficulty. In this paper, the topology and pipe diameter design optimization of the pipe network system are studied on the basis of ensuring that the pipeline is strictly laid along the road. In ESLP, the calculation target is the minimum path, so the edge weights in the connected graph needs to be converted from the pipe length to the annual cost, thus forming a minimum spanning tree problem with the annual cost as the weight. However, the cost is determined by the pipe length and pipe diameter, so "pipe diameter grading" was put forward. The grading was based on the relative position relation of different pipe segments, energy station and load points, and different "fuzzy pipe diameters" were assigned. After the first calculation, the "fuzzy pipe diameters" were corrected, and the iterative calculation was repeated to obtain the optimal solution. A case study showed that the pipe network optimization design for an energy system using the pipe diameter grading method was 1.88% more than the length of the pipe before optimization, but the annual cost was 2.38% less. Through further discussion, we found that the pipeline construction cost had the greatest impact on the total cost of the project. More importantly, the proposed design method was effective and valuable only if the different classifications of pipelines were sufficiently differentiated. Going back to the case study, this means that there was no increase in pipeline construction costs for DN100 of more than 33.1%, and no decrease for DN200 of more than 37.8%.

**Author Contributions:** Conceptualization, Y.R. and H.L. (Hongyun Liu); methodology, J.W. and H.L. (Hongyun Liu); software, H.L. (Hongyun Liu) and S.W.; validation, S.W., H.L. (Hongyun Liu) and J.Y.; formal analysis, H.L. (Hongyun Liu) and H.L. (Huiming Lu); investigation, H.L. (Huiming Lu), H.L. (Hongyun Liu) and J.Y.; resources, Y.R. and H.L. (Hongyun Liu); data curation, H.L. (Hongyun Liu) and H.L. (Huiming Lu); writing—original draft preparation, H.L. (Hongyun Liu), J.Y. and H.L. (Huiming Lu); writing—review and editing, J.W. and Y.R.; supervision, J.W.; project administration, Y.R. All authors have read and agreed to the published version of the manuscript.

**Funding:** This research was funded by the National Natural Science Foundation of China, grant number 5197081451.

**Institutional Review Board Statement:** Not applicable.

**Informed Consent Statement:** Not applicable.

**Data Availability Statement:** Not applicable.

**Acknowledgments:** The authors thank the anonymous referees for their valuable comments and suggestions.

**Conflicts of Interest:** The authors declare no conflict of interest.

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
