# Peer review of "A Novel Method for Network Design and Optimization of District Energy Systems: Considering Network Topology Planning and Pipe Diameter"

_applsci, doi:10.3390/app11041795_

Round 1

Reviewer 1 Report

Reviewer’s comments on manuscript applsci 1088476

This manuscript proposes and tests an innovative method for district heating system design with respect to road directions constraints. Such topic is relevant for Applied Sciences journal. Strong sides of this manuscript are introduction and research objectives formulation and justification. Weak sides: several important issues are omitted in Materials and Methods or need better explanation/justification. Results and discussion is too short, must be enriched at least by sensitivity analysis and comparison of the results with the findings of other relevant studies if possible. Conclusions can be rationalized. Abstract is unbalanced in favor of methods description.

Nevertheless, I expect that a significant manuscript quality can be achieved by major revision. Topic of the paper is interesting and if the authors believe in my comments and implement them during revision, a fine paper can be produced.

Detailed manuscript assessment and queries:

Abstract is unbalanced, providing much more space to methods description than to obtained results.

Language needs improvement and careful proofreading. See as an example line 15 “..PIPLINES that ARE installed along…” (missing verb + typing error) or line 38 “…which involves the integration of multidisciplinary.” (unfinished sentence). In addition to this:

What was meant with “energy station”? Maybe heating plant or combined heat and power unit? Or, “computing costs”? (line 202). There are several other sentences or expressions that are unclear. Authors need to involve a native speaker in the revision process, with extensive practice with technical English.

Introduction has sufficient scope and includes a suitable body of relevant and actual literature.  Authors should try to improve its structure thought, as the paragraphs are quite long and the key findings related to knowledge / method gap identification get lost. Study objectives and approach are presented in separate paragraph. Both objectives and method are meaningful and fit the scope of Applied Sciences journal.

Artwork quality is fine.

Materials and methods: Authors provide quite detailed description of the applied algorithms, accompanied by explanatory artwork which can be appreciated. Part 3 of the manuscript needs improvement:

  • please refer to all equations in text (applies to the whole manuscript)
  • please provide clear description how the considered system (case study) operates at part load. How (if) are variable hot water flows regulated (frequency converters, throttling valves…) and whether the hot water temperature changes with load (which is the usual case) and how does the water deltat change with load. All these factors influence the annual heat losses from pipelines and pumping costs and should be carefully considered in an optimization study
  • please state clearly that you considered the pipeline length both from energy source to load points and back to energy station when calculating all contributions to system annual cost.
  • Heat loss calculation: Insulation quality deteriorates over time and the heat losses increase. Especially in long-term projects, such as municipal heating network installation, real insulation performance should be considered, best by increasing the value of the considered insulation thermal conductivity.
  • Pumping costs: (equation (7)): please provide reference or explain the conversion factor (2.78exp(-7)). Please refer to the above question regarding possible hot water flow regulation as well. In addition: please explain (line 261): “If the power pump…is operated at full speed…” – is this really valid just for full load operation?
  • Table 1: Accuracy of design flow values is stunning. Are the authors sure they managed to estimate it with accuracy to tens of grams per hour? We always teach the students that it does not make any sense to write down values with bigger accuracy than is the uncertainty of their estimation.
  • Table 2: please provide reference for the input values stated in the left column. Considering the life time of the project as 20 years seems to short to me. Can it really be expected that the heating system will be completely redesigned after 20 years? I would expect at least 40 or 50 years.

Results and discussion part is too short and should be enriched. Sensitivity analysis is mandatory. Try to provide further results and discuss them thoroughly as to highlight the robustness and the value of the proposed and applied design methods. Moreover, a qualitative comparison with other relevant studies should be included as well.

Conclusions part should be rationalized. Try to reconsider and to shorten the first paragraph, it is too narrative.

Reviewer 2 Report

The paper deals with a network topology planning problem of a pipe network. The first part of the introduction contains a lot of information about the theoretical background of the work. It contains much information of similar works, however, this part just mentions only many times the applied optimization methodology, like convex optimization power flow or metaheuristic optimization techniques. It would be nice to mention the advantages/disadvantages of the proposed optimization methodology. For example, the convex optimization or mathematical programming-based techniques can guarantee that the found optimum is the global optimum, while most of the metaheuristics - evolutionary and genetic algorithms can't. This can be important to support the choice of the authors. Moreover, the no free lunch theorem of the mathematical optimization should be mentioned here, which states that the selection of these solvers is depends on the optimization task, as mentioned in the second section of this paper https://doi.org/10.3390/app10196653.

The novelty of the paper should be mentioned in the introduction, but I think this idea is only highlighted at the end of the methodology section.

Is it possible to write the pipeline classification in a more organized way? It starts from line 310 and maybe a picture or some change in the text can be more suitable for it.

Round 2

Reviewer 1 Report

Reviewer’s comments on the revised manuscript applsci 1088476

The revised manuscript improved visibly, both from formal and content point of view. I am satisfied with the majority of answers the authors provided to my comments as well as with the changes performed in the manuscript. I welcome the humorous side of the answers as well, it is rarely found in scientific communication and I enjoyed having one more student. :) I am glad the authors followed my recommendations and included the sensitivity analysis in the Results and Discussion section.

Following issues remained to be dealt with in a minor revision:

  • Referring to equations in text: There was a misunderstanding, maybe I did not express myself correctly in the recommendations for original submission improvement. By “referring to equations” I did not question, whether the equations are properly referenced. I just stated that each equation has to be referred to in plain text. As an example, line 141-142 should be modified as follows: “A spanning tree T0 is said to be a minimum spanning tree if, Equation (1):”.
  • Table 1 and elsewhere: please use “,” to separate thousands.
  • Language level has improved but still some issues remain. I think they can be dealt with in the language editing phase after manuscript acceptance for publishing.
  • References formatting: for all journal references, please include journal issue and pages/ paper number as well.
  • System part load operation description: I accept your answer; still it would be good to state the part load operation simulation and optimization as one of the future aims in part 4.3 of your manuscript.

Reviewer 2 Report

The authors answered all of my questions.
